# Towards Understanding The Effect of Loss Function on The Performance of Knowledge Graph Embedding

## Abstract

Knowledge graphs (KGs) represent world's facts in structured forms. KG completion exploits the existing facts in a KG to discover new ones. Translation-based embedding model (TransE) is a prominent formulation to do KG completion. Despite the efficiency of TransE in memory and time, it is claimed that TransE suffers from several *limitations* in encoding relation patterns (such as symmetric, reflexive relations etc). To solve that, most of the attempts have circled around the revision of the TransE *score function*, resulting more complicated score functions (such as TransA/D/G/H/R etc). These attempts have totally disregarded effect of loss functions to this end. We show that *loss functions* are key factors in this regard and disregarding them results in conclusions which are inaccurate or even wrong. More concretely, we show that the claimed limitations are inaccurate, as the effect of the loss was ignored. In this regard, we pose theoretical investigations of the main *limitations* of TransE in the light of *loss function*. To the best of our knowledge, so far, this has not been comprehensively investigated. We show that by a proper selection of the loss function for training the TransE model, the main limitations are mitigated. That is achieved by setting upper-bound for the scores of positive samples, defining the region of truth (i.e. the region that a triple is considered positive by the model). Our theoretical proofs with experimental results fill the gap in understanding of the limitation of translation-based class of embedding models and confirm the importance of the selection of loss functions for training the models and on their performance.

## 1 Introduction

Knowledge is considered as commonsense facts and other information accumulated from different sources. A Knowledge Graph (KG) is collection of facts and is usually represented as a set of triples $(h, r, t)$ where $h, t$ are entities and $r$ is a relation, e.g. $(iphone, hyponym, smartphone)$. Entities and relations are nodes and edges in the graph, respectively. As KGs are inherently incomplete, making prediction of missing links is a fundamental task in knowlege graph analyses. Among different approaches used for KG completion, KG Embedding (KGE) has recently received growing attentions. KGE embeds entities and relations as low dimensional vectors known as embeddings. To measure the degree of plausibility of a triple, a scoring function is defined over the embeddings.

TransE, Translation-based Embedding model, (Bordes et al., 2013) is one of the most widely used KGE models. The original assumption of TransE is to hold: $\mathbf{h} + \mathbf{r} = \mathbf{t}$, for every positive triple $(h, r, t)$ where $\mathbf{h}, \mathbf{r}, \mathbf{t} \in \mathbb{R}^d$ are embedding vectors of head ($h$), relation ($r$) and tail ($t$) respectively. TransE and its many variants like TransH (Wang et al., 2014) and TransR (Lin et al., 2015b), underperform greatly compared to the current state-of-the-art models. That is reported to be due to the limitations of their scoring functions. For instance, (Wang et al., 2018) reports that TransE cannot encode a relation pattern which is neither reflexive nor irreflexive.

In most of these works the effect of the loss function is ignored and the provided proofs are based on the assumptions that are not fulfilled by the associated loss functions. For instance (Sun et al., 2019) proves that TransE is incapable of encoding symmetric relation. To this end the loss function must enforce the distance of $\|\mathbf{h} + \mathbf{r} - \mathbf{t}\|$ to zero, but this is never fulfilled (or even *approximated*) by the employed loss function. Similarly, (Wang et al., 2018) reports that TransE cannot encode a relation pattern which is neither reflexive nor irreflexive and (Wang et al., 2014) adds that TransE cannot

properly encode reflexive, one-to-many, many-to-one and many-to-many relations. However, as mentioned earlier, such reported limitations are not accurate and the problem is not fully investigated due to the effect of the loss function.

In this regards, although TransH, TransR and TransD (Wang et al., 2014; Lin et al., 2015b; Ji et al., 2015) addressed the reported problem of TransE in one-to-many, many-to-one, many-to-many and reflexive etc, they were misled by the assumption (enforcing $\|\mathbf{h} + \mathbf{r} - \mathbf{t}\|$ to be zero) that was not fulfilled by the employed loss function. Considering the same assumption, (Kazemi & Poole, 2018) investigated three additional limitations of TransE, FTransE (Feng et al., 2016), STransE (Nguyen et al., 2016), TransH and TransR models: (i) if the models encode a reflexive relation $r$, they automatically encode symmetric, (ii) if the models encode a reflexive relation $r$, they automatically encode transitive and, (iii) if entity $e_1$ has relation $r$ with every entity in $\Delta \in \mathcal{E}$ and entity $e_2$ has relation $r$ with one of entities in $\Delta$, then $e_2$ must have the relation $r$ with every entity in $\Delta$.

Assuming that the loss function enforces the norm to be zero, the aforementioned works have investigated these limitations by focusing on the capability of *scoring* functions in encoding relation patterns. However, we prove that the selection of *loss* function affects the boundary of score functions; consequently, the selection of loss functions significantly affects the limitations. Therefore, the above mentioned theories corresponding to the limitations of translation-based embedding models in encoding relation patterns are inaccurate. We pose new theories about the limitations of TransX(X=H,D,R, etc) models considering the *loss* functions. To the best of our knowledge, it is the first time that the effect of loss function is investigated to prove theories corresponding to the limitations of translation-based models.

In a nutshell, the key contributions of this paper is as follows. (i) We show that different loss functions enforce different upper-bounds and lower-bounds for the scores of positive and negative samples respectively. This implies that existing theories corresponding the limitation of TransX models are inaccurate because the effect of loss function is ignored. We introduce new theories accordingly and prove that the proper selection of loss functions mitigates the main limitations. (ii) We reformulate the existing loss functions and their optimization problems as an standard constrained optimization problem. This makes perfectly clear that how each of the loss functions affect on the boundary of triples scores and consequently ability of relation pattern encoding. (iii) Using symmetric relation patterns, we obtain a proper upper-bound of positive triples score to enable encoding of symmetric patterns. (iv) We additionally investigate the theoretical capability of translation-based embedding model when translation is applied in the complex space (TransComplEx). We show that TransComplEx is a more powerful embedding model with fewer theoretical limitations in encoding different relation patterns such as symmetric while it is efficient in memory and time.

## 2 RELATED WORKS

Most of the previous work have investigated the capability of translation-based class of embedding models considering solely the formulation of the score function. Accordingly, in this section, we review the score functions of TransE and some of its variants together with their capabilities. Then, in the next section the existing limitations of Translation-based embedding models emphasized in recent works are reviewed. These limitations will be reinvestigated in the light of *score* and *loss* functions in the section 4.

The score of **TransE** (Bordes et al., 2013) is defined as: $f_r(h, t) = \|\mathbf{h} + \mathbf{r} - \mathbf{t}\|$. **TransH** (Wang et al., 2014) projects each entity ($\mathbf{e}$) to the relation space ($\mathbf{e}_\perp = \mathbf{e} - \mathbf{w}_r \mathbf{e} \mathbf{w}_r^T$). The score function is defined as $f_r(h, t) = \|\mathbf{h}_\perp + \mathbf{r} - \mathbf{t}_\perp\|$. TransH can encode reflexive, one-to-many, many-to-one and many-to-many relations. However, recent theories (Kazemi & Poole, 2018) prove that encoding reflexive results in encoding the both symmetric and transitive which is undesired. **TransR** (Lin et al., 2015b) projects each entity ($\mathbf{e}$) to the relation space by using a matrix provided for each relation ($\mathbf{e}_\perp = \mathbf{e} \mathbf{M}_r$, $\mathbf{M}_r \in \mathbb{R}^{d_e \times d_r}$). TransR uses the same scoring function as TransH. **TransD** (Ji et al., 2015) provides two vectors for each individual entities and relations ($\mathbf{h}, \mathbf{h}_p, \mathbf{r}, \mathbf{r}_p, \mathbf{t}, \mathbf{t}_p$). Head and tail entities are projected by using the following matrices: $\mathbf{M}_{rh} = \mathbf{r}_p^T \mathbf{h}_p + \mathbf{I}^{m \times n}, \mathbf{M}_{rt} = \mathbf{r}_p^T \mathbf{t}_p + \mathbf{I}^{m \times n}$. The score function of TransD is similar to the score function of TransH.

**RotatE** (Sun et al., 2019) rotates the head to the tail entity by using relation. RotatE embeds entities and relations in the Complex space. By inclusion of constraints on the norm of entity vectors, the

model would be degenerated to TransE. The scoring function of RotatE is $f_r(h, t) = \|\mathbf{h} \circ \mathbf{r} - \mathbf{t}\|$, where $\mathbf{h}, \mathbf{r}, \mathbf{t} \in \mathbb{C}^d$, and $\circ$ is element-wise product. RotatE obtains the state-of-the-art results using very big embedding dimension (1000) and a lot of negative samples (1000). **TorusE** (Ebisu & Ichise, 2018) fixes the problem of regularization in TransE by applying translation on a compact Lie group. The model has several variants including mapping from torus to Complex space. In this case, the model is regarded as a very special case of RotatE Sun et al. (2019) that applies rotation instead of translation in the target the Complex space. According to Sun et al. (2019), TorusE is not defined on the entire Complex space. Therefore, it has less representation capacity. TorusE needs a very big embedding dimension (10000 as reported in Ebisu & Ichise (2018)) which is a limitation.

## 3 THE MAIN LIMITATIONS OF TRANSLATION-BASED EMBEDDING MODELS

We review six limitations of translation-based embedding models in encoding relation patterns (e.g. reflexive, symmetric) mentioned in the literature (Wang et al., 2014; Kazemi & Poole, 2018; Wang et al., 2018; Sun et al., 2019).

**Limitation L1**. TransE cannot encode reflexive relations when the relation vector is non-zero (Wang et al., 2014).

**Limitation L2**. TransE cannot encode a relation which is neither reflexive nor irreflexive. To see that, if TransE encodes a relation $r$, which is neither reflexive nor irreflexive we have $\mathbf{h}_1 + \mathbf{r} = \mathbf{h}_1$ and $\mathbf{h}_2 + \mathbf{r} \neq \mathbf{h}_2$, resulting $\mathbf{r} = \mathbf{0}$, $\mathbf{r} \neq \mathbf{0}$ which is a contradiction (Wang et al., 2018).

**Limitation L3**. TransE cannot properly encode symmetric relation when $\mathbf{r} \neq \mathbf{0}$. To see that (Sun et al., 2019), if $r$ is symmetric, then we have: $\mathbf{h} + \mathbf{r} = \mathbf{t}$ and $\mathbf{t} + \mathbf{r} = \mathbf{h}$. Therefore, $\mathbf{r} = \mathbf{0}$ and so all entities appeared in head or tail parts of training triples will have the same embedding vectors.

The following limitations hold for TransE, FTransE, STransE, TransH and TransR (Feng et al., 2016; Nguyen et al., 2016; Kazemi & Poole, 2018):

**Limitation L4**. If $r$ is reflexive on $\Delta \in \mathcal{E}$, where $\mathcal{E}$ is the set of all entities in the KG, then $r$ must also be symmetric.

**Limitation L5**. If $r$ is reflexive on $\Delta \in \mathcal{E}$, $r$ must also be transitive.

**Limitation L6**. If entity $e_1$ has relation $r$ with every entity in $\Delta \in \mathcal{E}$ and entity $e_2$ has relation $r$ with one of entities in $\Delta$, then $e_2$ must have the relation $r$ with every entity in $\Delta$.

## 4 OUR MODEL

TransE and its variants underperform compared to other embedding models due to their limitations we iterated in Section 3. In this section, we reinvestigate the limitations. We show that the corresponding theoretical proofs are inaccurate because the effect of loss function is ignored. So we propose new theories and prove that each of the limitations of TransE are resolved by revising either the *scoring* function or the *loss*. In this regard, we consider several loss functions and their effects on the boundary of the TransE scoring function. For each of the loss functions, we pose theories corresponding to the limitations. We additionally investigate the limitations of TransE using each of the loss functions while translation is performed in the complex space and show that by this new approach the aforementioned limitations are lifted. Our new model, TransComplEx, with a proper selection of loss function addresses the above problems.

### 4.1 TRANSCOMPLEX: TRANSLATIONAL EMBEDDING MODEL IN THE COMPLEX SPACE

TransComplEx translates head entity vector to the conjugate of the tail vector using relation in the complex space (Trouillon et al., 2016). Assuming $\mathbf{h}, \mathbf{r}, \mathbf{t} \in \mathbb{C}^d$ be complex vectors of dimension $d$, the score function is defined as follows:

$$f_r(h, t) = \|\mathbf{h} + \mathbf{r} - \bar{\mathbf{t}}\|$$

**Advantages of TransComplEx:** We highlight four advantages of using the above formulation. (i) Comparing to TransE and its variants, TransComplEx has less limitations in encoding different relation patterns. The theories and proofs are provided in the next part. (ii) Using conjugate of tail

vector in the formulation enables the model to make difference between the role of an entity as subject or object. This cannot be properly captured by TransE and its variants. (iii) Given the example $(A, Like, Juventus)$, $(Juventus, hasPlayer, C.Ronaldo)$, that $C.Ronaldo$ plays for $Juventus$ may affect the person $A$ to like the team. This type of information cannot be properly captured by models such as CP decomposition (Hitchcock, 1927) where two independent vectors are provided (Kazemi & Poole, 2018) for $Juventus$ (for subject and object). In contrast, our model uses same real and imaginary vectors for $Juventus$ when it is used as subject or object. Therefore, TransComplEx can properly capture dependency between the two triples with the same entity used as subject and object. And finally, (iiii) ComplEx (Trouillon et al., 2016) has much more computational complexity comparing to TransComplEx because it needs to compute eight vector multiplications to obtain score of a triple while our model only needs to do four vector summation/subtractions. In the experiment section, we show that TransComplEx outperforms ComplEx on various dataset.

## 4.2 REINVESTIGATION OF THE LIMITATIONS OF TRANSLATION-BASED MODELS

The aim of this part is to analyze the limitations of Translation-based embedding models (including TransE and TransComplEx) by considering the effect of both *score* and *loss* functions. Different loss functions provide different upper-bound and lower-bound for positive and negative triples scores, respectively. Therefore, the loss functions affect the limitations of the models to encode relation patterns. In this regard, the existing works consider a positive triple of $(h, r, t)$ and a negative triple of $(h', r, t')$ to satisfy some assumptions in a score function. For instance in TransE where $f_r(h, t) = \|\mathbf{h} + \mathbf{r} - \mathbf{t}\|$, it is expected that $f_r(h, t) = 0$ and $f_r(h', t') > 0$. Unfortunately, as we show later, this can not be fulfilled (or even approximated) by using the proposed loss functions (e.g. margin ranking loss and RotatE loss).

To investigate and address the limitations, we propose four conditions (Table-1) for taking a triple as positive or negative by the score function. This is done by defining upper-bound and lower-bound for the scores. We show that these conditions can be approximated by proper loss functions, and in this regards we propose four losses that each will handle one of the condition.

Table 1: Region of truth and falsity.

| Condition | Positive | Negative | $\gamma_1, \gamma_2 \in \mathbb{R}$ |
|---|---|---|---|
| (a) | $f_r(h, t) = \gamma_1,$ | $f_r(h', t') \geq \gamma_2$ | $\gamma_1 = 0, \gamma_2 > 0$ |
| (b) | $f_r(h, t) = \gamma_1$ | $f_r(h', t') \geq \gamma_2$ | $\gamma_2 > \gamma_1 > 0$ |
| (c) | $f_r(h, t) \leq \gamma_1$ | $f_r(h', t') \geq \gamma_2$ | $\gamma_2 > \gamma_1 > 0$ |
| (d) | $f_r(h, t) \leq \gamma_{1(h,r,t)}$ | $f_r(h', t') \geq \gamma_{2(h,r,t)}$ | $\gamma_{2(h,r,t)} > \gamma_{1(h,r,t)} > 0$ |

To better comprehend Table-1, we have visualized the conditions in Figure 1. The condition (a) indicates a triple is positive if $\mathbf{h} + \mathbf{r} = \mathbf{t}$ holds. It means that the length of *residual vector* i.e. $\epsilon = \mathbf{h} + \mathbf{r} - \mathbf{t}$, is zero. It is the most strict condition that expresses being positive. Authors in (Sun et al., 2019; Kazemi & Poole, 2018) consider this condition to prove their theories as well as limitation of TransE in encoding of symmetric relations. However, the employed loss cannot approximate (a), rather it fulfills (c), resulting the reported limitation to be void in that setting.

Condition (b) considers a triple to be positive if its residual vector lies on a hyper-sphere with radius $\gamma_1$. It is less restrictive than (a) which considers a point to express being positive. The optimization problem that approximates the conditions (a) ($\gamma_1 = 0$) and (b) ($\gamma_1 > 0$) is as follows:

$$\begin{cases} \min_{\xi_{h,t}} \sum_{(h,r,t) \in S^+} \xi_{h,t}^2 \\ f_r(h, t) = \gamma_1, \ (h,r,t) \in S^+ \\ f_r(h', t') \geq \gamma_2 - \xi_{h,t}, \ (h',r,t') \in S^- \\ \xi_{h,t} \geq 0 \end{cases} \quad (1)$$

where $S^+, S^-$ are the set of positive and negative samples respectively. $\xi_{h,t}$ are slack variables to reduce the effect of noise in negative samples (Nayyeri et al., 2019).

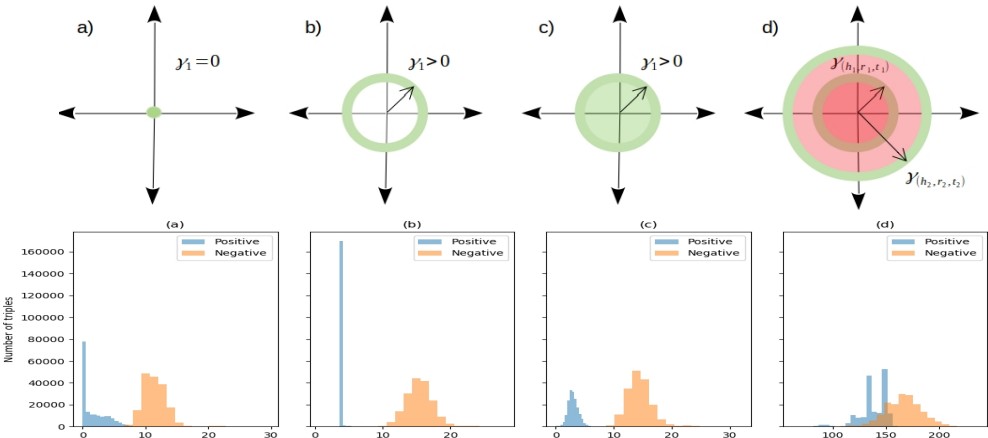

Figure 1: **Top:** Visualization of truth region (positive) of a triple according to Table 1. The residual vector $\epsilon$, (a) becomes **0**, (b) lies on the border of a sphere with radius $\gamma_1$, (c) lies inside of a sphere with radius $\gamma_1$, and (d) $\epsilon_{(h_1,r_1,t_1)}$ lies inside of a sphere with radius $\gamma_{(h_1,r_1,t_1)}$. **Bottom:** The histogram of the scores of triples when TransE is trained on WordNet (WN18RR) using the losses of Equation 2 ($\gamma_1 = 0$), 2 ($\gamma_1 = 4$), 4 ($\gamma_1 = 4$) and 5 ($\gamma = 6$) respectively. Each of the bottom figures is the approximation of the corresponding conditions (a) to (d).

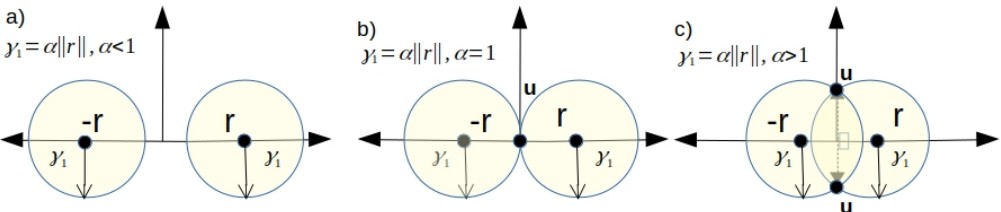

Figure 2: Necessity condition for encoding symmetric relation: (a) when $\alpha < 1$, the model cannot encode it. (b) when $\alpha = 1$, the intersection of two hyperspheres is a point. **u = 0** means embedding vectors of all entities should be same. Therefore, symmetric cannot be encoded. (c) when $\alpha > 1$, symmetric can be encoded as there are more than one point in the intersection of two hyperspheres.

One loss function that approximates the conditions (a) and (b) is as follows. Please note that for case (a) we set $\gamma_1 = 0$ and for case (b) we set $\gamma_1 > 0$ in the formula.

$$\mathcal{L}_{a|b} = \sum_{(h,r,t) \in S^+} \left( \lambda_1 \|f_r(h,t) - \gamma_1\| + \sum_{(h',r,t') \in S^-_{(h,r,t)}} \lambda_2 \max(\gamma_2 - f_r(h',t'), 0) \right). \quad (2)$$

Condition (c) considers a triple to be positive if its residual vector lies inside a hyper-sphere of radius $\gamma_1$. The optimization problem that approximates the condition (c) is as follows (Nayyeri et al., 2019):

$$\begin{cases} \min_{\xi_{h,t}} \sum_{(h,r,t) \in S^+} \xi_{h,t}^2 \\ f_r(h,t) \leq \gamma_1, \ (h,r,t) \in S^+ \\ f_r(h',t') \geq \gamma_2 - \xi_{h,t}, \ (h',r,t') \in S^- \\ \xi_{h,t} \geq 0 \end{cases} \quad (3)$$

The loss function that approximates the condition (c) is as follows Nayyeri et al. (2019):

$$\mathcal{L}_c = \sum_{(h,r,t) \in S^+} \left( \lambda_1 \max(f_r(h,t) - \gamma_1, 0) + \sum_{(h',r,t') \in S^-_{(h,r,t)}} \lambda_2 \max(\gamma_2 - f_r(h',t'), 0) \right). \quad (4)$$

**Remark:** The loss function which is defined in Zhou et al. (2017b) is slightly different from the loss in 4. The former slides the margin while the latter fixes the margin by inclusion of a lower-bound for the score of negative triples. Both losses put an upper-bound for scores of positive triples.

Apart from the loss 4, the RotatE loss Sun et al. (2019) also approximates the condition (c). The formulation of the RotatE loss is as follows:

$$\mathcal{L}_c^{RotatE} = -\sum_{(h,r,t)\in S^+} \Big( \log \sigma(\gamma - f_r(h,t)) + \sum_{(h',r,t')\in S^-_{(h,r,t)}} \log \sigma(f_r(h',t') - \gamma) \Big).$$

Condition (d) is similar to (c), but provides different $\gamma_1, \gamma_2$ for each triples. Using (d), there is not a unique region of truth for all positive triples, rather for each positive triple $(h, r, t)$ and its corresponding negative of $(h', r, t')$ there are triple-specific region of truth and falsity. Margin ranking loss (Bordes et al., 2013) approximates (d). Defining $[x]_+ = \max(0, x)$, the loss is defined as:

$$\mathcal{L}_d = \sum \sum [f_r(h,t) + \gamma - f_r(h',t')]_+ . \tag{5}$$

To investigate the limitations, we must assume that the relation vectors is not null otherwise we will have the same embedding for head and tail which is undesirable. Considering the conditions (a) to (d), we investigate the limitations L1 to L6 and we prove that existing theories are just valid under (a), which is not fulfilled under the given loss. In this regard we have the following theorem. For complete proofs, please refer to the appendix of the paper.

**Theorem T1.** *(Addressing L1)*: TransE and TransComplEx cannot infer a reflexive relation pattern with a non-zero relation vector under (a). However, under (b-d), TransE and TransComplEx can infer reflexive pattern.

**Theorem T2.** *(Addressing L2)*: (i) TransComplEx can infer a relation which is neither reflexive nor irreflexive under (b-d). (ii) TransE cannot infer a relation which is neither reflexive nor irreflexive under (a-d).

**Theorem T3.** *(Addressing L3)*: (i) TransComplEx can infer symmetric relations under (a-d). (ii) TransE cannot infer symmetric relations under (a) with non-zero vector for relation. (iii) TransE can infer a symmetric relation under (b-d).

*Proof:* Proofs of (i) and (ii) are provided in the appendix. For (iii) we have:

Under (b), for TransE we have $\|\mathbf{h} + \mathbf{r} - \mathbf{t}\| = \gamma_1$ and $\|\mathbf{t} + \mathbf{r} - \mathbf{h}\| = \gamma_1$. The necessity condition for encoding symmetric relation is $\|\mathbf{h}+\mathbf{r}-\mathbf{t}\| = \|\mathbf{t}+\mathbf{r}-\mathbf{h}\|$. This implies $\|\mathbf{h}\| \cos(\theta_{h,r}) = \|\mathbf{t}\| \cos(\theta_{t,r})$. Let $\mathbf{h} - \mathbf{t} = \mathbf{u}$, by definition we have $\|\mathbf{u} + \mathbf{r}\| = \gamma_1$, $\|\mathbf{u} - \mathbf{r}\| = \gamma_1$. Now let $\gamma_1 = \alpha\|r\|$, we have:

$$\begin{cases} \|\mathbf{u}\|^2 + (1-\alpha^2)\|\mathbf{r}\|^2 = -2\langle\mathbf{u},\mathbf{r}\rangle \\ \|\mathbf{u}\|^2 + (1-\alpha^2)\|\mathbf{r}\|^2 = 2\langle\mathbf{u},\mathbf{r}\rangle \end{cases} \tag{6}$$

Therefore we have: $\|\mathbf{u}\|^2+(1-\alpha^2)\|\mathbf{r}\|^2 = -(\|\mathbf{u}\|^2+(1-\alpha^2)\|\mathbf{r}\|^2)$, which can be written as $\|\mathbf{u}\|^2 = (\alpha^2 - 1)\|\mathbf{r}\|^2$. To avoid contradiction we must have $\alpha > 1$. Once $\alpha > 1$ we have $\cos(\theta_{u,r}) = \pi/2$. Therefore, TransE can encode symmetric relation with condition (b), when $\gamma_1 = \alpha\|r\|$ and $\alpha > 1$. Figure 2 shows different conditions for encoding symmetric relation.

Conditions (c-d) are directly resulted from (b), as it is subsumed by (c) and (d). That completes the proof.

**Theorem T4.** *(Addressing L4)*: For both TransE and TransComplEx, (i) Limitation L4 holds under (a). (ii) Limitation L4 is not valid under (b-d).

**Theorem T5.** *(Addressing L5)*: For both TransE and TransComplEx, (i) Limitation L5 holds under (a). (ii) Limitation L5 holds is not valid under (b-d).

**Theorem T6.** *(Addressing L6)*: For both TransE and TransComplEx, (i) Limitation L6 holds under (a). (ii) Limitation L6 is not valid under (b-d).

### 4.3 ENCODING RELATION PATTERNS IN TRANSCOMPLEX

Most of KGE models learn from triples. Recent work incorporates relation patterns such as transitive, symmetric on the top of triples to further improve performance of models. For instance, ComplEx-NNE+AER (Ding et al., 2018) encodes implication relation in the ComplEx model. RUGE (Guo et al., 2018) injects First Order Horn Clause rules in an embedding model. SimplE

(Kazemi & Poole, 2018) captures symmetric, antisymmetric and inverse relations by weight tying in the model. Inspired by (Minervini et al., 2017) and considering the score function of TransComplEx, in this part, we derive formulae for equivalence, symmetric, inverse and implication to be used as regularization terms in the optimization problem. Therefore, TransComplEx incorporates different relation patterns to optimize the embeddings.

**Symmetric:** Assume that $r$ is a symmetric relation. To encode it we should have $f_r(h, t) \approx f_r(t, h)$, therefore $\|f_r(h, t) - f_r(t, h)\| = 0$. According to the definition of score function of TransComplEx, we have the following algebraic formulae: $\mathcal{R}_{\text{Sym}} := \|Re(\mathbf{h}) - Re(\mathbf{t})\| = 0$.

Using similar argument to symmetric, the following formulae are derived for transitive, composition, inverse and implication.

**Equivalence:** Let $p, q$ be equivalence relations, therefore we should have $f_p(h, t) \approx f_q(h, t)$. We obtain, $\mathcal{R}_{\text{Eq}} := \|\mathbf{p} - \mathbf{q}\| = 0$.

**Implication:** Let $p \to q$, be the implication rule. We obtain $\mathcal{R}_{\text{Imp}} := \max(f_p(h, t) - f_q(h, t), 0) = 0$.

**Inverse:** Let $r \longleftrightarrow r^{-1}$ be the inverse relation. We obtain $\mathcal{R}_{\text{Inv}} := \|\mathbf{r} - \mathbf{r}^{-1}\|$.

Finally, the following optimization problem should be solved:

$$\min_{\theta} \; \mathcal{L} + \sum \eta_i \mathcal{R}_i \tag{7}$$

where $\theta$ is embedding parameters, $\mathcal{L}$ is one of the losses 2, 4 or 5 and $\mathcal{R}_i$ is one of the derived formulae mentioned above.

## 5 EXPERIMENTS AND EVALUATIONS

In this section, we evaluate performance of our model, TransComplEx, with different loss functions on the link prediction task. The aim of the task is to complete the triple $(h, r, ?)$ or $(?, r, t)$ by prediction of the missed entity $h$ or $t$. Filtered Mean Rank (MR), Mean Reciprocal Rank (MRR) and Hit@10 are used for evaluations (Wang et al., 2017; Lin et al., 2015b).

**Dataset.** We use two dataset extracted from Freebase (Bollacker et al., 2008) (i.e. FB15K (Bordes et al., 2013) and FB15K-237 (Toutanova & Chen, 2015)) and two others extracted from Word-Net (Miller, 1995) (i.e. WN18 (Bordes et al., 2013) and WN18RR (Dettmers et al., 2018)). FB15K and WN18 are earlier dataset which have been extensively used to compare performance of KGEs. FB15K-237 and WN18RR are two dataset which are supposed to be more challenging after removing inverse patterns from FB15K and WN18. Guo et al. (2018) and Ding et al. (2018) extracted different relation patterns from FB15K and WN18 respectively. The relation patterns are provided by their confidence level, e.g. $(a, BornIn, b) \xrightarrow{0.9} (a, Nationality, b)$. We drop the relation patterns with confidence level less than 0.8. Generally, we use 454 and 14 relation patterns for FB15K and WN18 respectively. We do grounding for symmetric and transitive relation patterns. Thanks to the formulation of score function, grounding is not needed for inverse, implication and equivalence.

**Experimental Setup.** We implement TransComplEx with the losses 2, 4 and 5 and TransE with the loss 4 in PyTorch. Adagrad is used as an optimizer. We generate 100 mini-batches in each iteration. The hyperparameter corresponding to the score function is embedding dimension $d$. We add slack variables to the losses 2 and 4 to have soft margin as in (Nayyeri et al., 2019). The loss 4 is rewritten as follows Nayyeri et al. (2019):

$$\min_{\xi_{h,t}^r} \sum_{(h,r,t)\in S^+} \left( \lambda_0 {\xi_{h,t}^r}^2 + \lambda_1 \max(f_r(h,t) - \gamma_1, 0) + \right.$$
$$\left. \lambda_2 \sum_{(h,r,t)\in S_{h',r,t'}^-} \max(\gamma_2 - f_r(h^{'}, t^{'}) - \xi_{h,t}^r, 0) \right). \tag{8}$$

We set $\lambda_1$ and $\lambda_2$ to one and search for the hyperparameters $\gamma_1 (\gamma_2 > \gamma_1)$ and $\lambda_0$ in the sets $\{0.1, 0.2, \dots, 2\}$ and $\{0.01, 0.1, 1, 10, 100\}$ respectively. Moreover, we generate $\alpha \in \{1, 2, 5, 10\}$

negative samples per each positive. The embedding dimension and learning rate are tuned from the sets $\{100, 200\}, \{0.0001, 0.0005, 0.001, 0.005, 0.01\}$ respectively. All hyperparameters are adjusted by early stopping on validation set according to MRR. RPTransComplEx# denotes the TransComplEx model which is trained by the loss function # (2, 4, 5). RP indicates that relation patterns are injected during learning by regularizing the derived formulae (see 7). TransComplEx# refers to our model trained with the loss # without regularizing relation patterns formulae. The same notation is used for TransE#. The optimal configurations for RPTransComplEx are provided in the appendix.

| | FB15k | | | WN18 | | |
|---|---|---|---|---|---|---|
| | MR | MRR | Hits@10 | MR | MRR | Hits@10 |
| TransE (Bordes et al., 2013) | 125 | - | 47.1 | 251 | - | 89.2 |
| TransH (bern) (Wang et al., 2014)* | 87 | - | 64.4 | 388 | - | 82.3 |
| TransR (bern) (Lin et al., 2015b)* | 77 | - | 68.7 | 225 | - | 92.0 |
| TransD (bern) (Ji et al., 2015)* | 91 | - | 77.3 | **212** | - | 92.2 |
| TransE-RS (bern) (Zhou et al., 2017a)* | 63 | - | 72.1 | 371 | - | 93.7 |
| TransH-RS (bern) (Zhou et al., 2017a)* | 77 | - | 75.0 | 357 | - | 94.5 |
| TorusE (Ebisu & Ichise, 2019) | - | 73.3 | 83.2 | - | 94.7 | 95.4 |
| TorusE(with WNP) (Ebisu & Ichise, 2019) | - | **75.1** | 83.5 | - | **94.7** | 95.4 |
| R-GCN (Schlichtkrull et al., 2018)+ | - | 65.1 | 82.5 | - | 81.4 | **95.5** |
| ConvE (Dettmers et al., 2018)++ | 51 | 68.9 | 85.1 | 504 | 94.2 | **95.5** |
| ComplEx (Trouillon et al., 2016)++ | 106 | 67.5 | 82.6 | 543 | 94.1 | 94.7 |
| ANALOGY (Liu et al., 2017)++ | 121 | 72.2 | 84.3 | - | 94.2 | 94.7 |
| RotatE (Sun et al., 2019) | 48 | 69.0 | 86.1 | 433 | **94.8** | 95.5 |
| SimplE (Kazemi & Poole, 2018) | - | 72.7 | 83.8 | - | 94.2 | 94.7 |
| SimplE+ (Fatemi et al., 2018) | - | 72.5 | 84.1 | - | 93.7 | 93.9 |
| PTransE (Lin et al., 2015a) | 58 | - | 84.6 | - | - | - |
| KALE (Guo et al., 2016) | 73 | 52.3 | 76.2 | 241 | 53.2 | 94.4 |
| RUGE (Guo et al., 2018) | 97 | 76.8 | 86.5 | - | - | - |
| ComplEx-NNE+AER (Ding et al., 2018) | 116 | 80.3 | 87.4 | 450 | 94.3 | 94.8 |
| RPTransComplEx2 | **38** | 70.5 | 88.3 | 451 | 92.7 | 94.8 |
| RPTransComplEx4 | **38** | 72.4 | **88.8** | 275 | 92.4 | 95.4 |
| RPTransComplEx5 | 59 | 61.7 | 82.2 | 547 | 94.0 | 94.7 |
| TransComplEx4 | **38** | 68.2 | 87.5 | 284 | 92.2 | **95.5** |
| TransE4 | 46 | 64.8 | 87.2 | 703 | 68.7 | 94.5 |

Table 2: Link prediction results. Rows 1-8: Translation-based models with no injected relation patterns. Rows 9-13: basic models with no injected relation patterns. Rows 14-18: models which encode relation patterns. Results labeled with *, + and ++ are taken from (Zhou et al., 2017a), (Ebisu & Ichise, 2019) and (Akrami et al., 2018) respectively, while the rest are taken from original papers/code. Dashes: results could not be obtained.

**Results.** Table 2 presents comparison of TransComplEx and its relation pattern encoded variants (RPTransComplEx) with three classes of embedding models on the most famous two datasets of FB15K and WN18. The first category (CAT1) of models consist of translation-based model (e.g. TransX, TorusE). The second category (CAT2) are embedding models which are not translation-based (e.g. ConvE, ComplEx, ANALOGY). The previous two categories (CAT1/2) can learn relation patterns that are reflected in triples. In other words they learn patterns implicitly from existing triples. The last category (CAT3) consist of models which can explicitly encode (inject) rules in their training process (e.g. RUGE, ComplEx-NNE+AER, SimplE, SimplE+). The models in CAT3 are actually trained on relation patterns as well as triples.

We trained relation pattern version of our model i.e. RPTransComplEx using the losses of 2, 4, 5. All variants of RPTransComplEx, except PRTransComplEx4, are generally performing better than models in CAT3 which also explicitly inject relation patterns. The exception, PRTransComplEx5, is due to the fact that it uses margin ranking loss which we already showed its restriction regarding condition (d). In this regard, please consider histogram (d) in Figure 1.

As all limitations were mitigated by condition (c), we expected that RPTransComplEx4, which is associated to (c), should perform better than others. This is empirically approved as shown in Table-

| | FB15k-237 | | | WN18RR | | |
|---|---|---|---|---|---|---|
| | MR | MRR | Hits@10 | MR | MRR | Hits@10 |
| TransE (Bordes et al., 2013) (our implementation) | **205** | 27.1 | 45.2 | **3806** | 19.5 | 45.1 |
| DistMult (Bordes et al., 2013)+ | - | 24.1 | 41.9 | - | 43.0 | 49.0 |
| ComplEx (Trouillon et al., 2016)+ | - | 24.0 | 41.9 | - | 44.0 | 51.0 |
| R-GCN (Schlichtkrull et al., 2018)+ | - | 24.8 | 41.7 | - | - | - |
| ConvE (Dettmers et al., 2018)+ | - | 31.6 | 49.1 | - | 46.0 | 48.0 |
| TorusE (Ebisu & Ichise, 2019) | - | 30.5 | 48.4 | - | 45.2 | 51.2 |
| TorusE (with WNP) (Ebisu & Ichise, 2019) | - | 30.7 | 48.5 | - | 46.0 | 53.4 |
| RotatE (Sun et al., 2019) | 211 | 31.1 | 49.4 | 4789 | **47.3** | **54.9** |
| RPTransComplEx2 | 210 | 27.7 | 46.4 | - | - | - |
| RPTransComplEx4 | 226 | **31.9** | **49.5** | - | - | - |
| RPTransComplEx5 | 216 | 25.3 | 43.8 | - | - | - |
| TransComplEx4 | 223 | 31.7 | 49.3 | 4081 | 38.9 | 49.8 |
| TransE4 | **205** | 27.2 | 45.3 | 3850 | 20.0 | 47.5 |

Table 3: Link prediction results. Rows 1-8: basic models with no injected relation patterns. Results labeled with + are taken from (Ebisu & Ichise, 2019) while the rest are taken from original papers/code. Dashes: results could not be obtained.

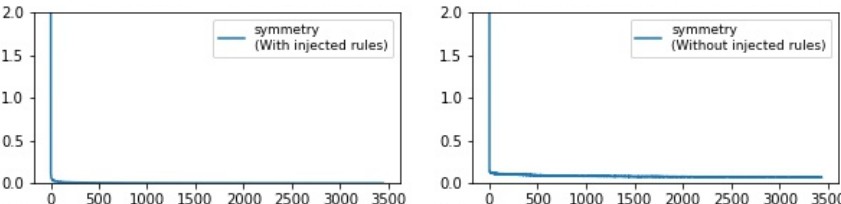

Figure 3: The convergence of symmetric relation loss on FB15K

2. The question is that how our model behaves under the loss 4 while we additionally disregard injection of relation patterns. To this end, we considered TransComplEx4 which uses loss 4 but does not do any injection. Since the performances of RPTransComplEx4 and TransComplEx4 are very close, we conclude that the latter also learns the pattern over the existing triples quite efficiently. This is also confirmed by the convergence performance of them as shown in Figure 3 over the symmetric relation.

To have a fair comparison[1] with the categories of models which disregard injection of relation patterns (CAT1/2) we used the TransComplEx4 version of our approach. As shown in Table-2, we can observe that TransComplEx4 performs better than other models in CAT1/2 considering MR and Hits@10 and performing very closely to the best models on MRR.

As discussed earlier, FB15K-237 and WN18RR are two more challenging dataset provided recently. Table 3 presents the comparisons of our models with those other models that their performance results were available on these two datasets. Similar to our previous discussion on Table-2, we observe that PRTransComplEx4 and TransComplEx4 are performing closely which means that the latter has learned the relation pattern over triples very well without any injection. On WN18RR, TorusE was performing better than TransComplEx4 due to big embedding dimension of 10,000. On FB15k-237, RPTransComplEx4 performed better that others on MRR and Hits@10.

In order to investigate the effect of grounding, we train RPTransComplEx4 in two settings: 1) RPTransComplEx4(w grounding) is trained when the grounded patterns are injected, 2) RPTransComplEx4(w/o grounding) is trained when the relation patterns which are not grounded used. According to the Table 4, the grounding does not affect the performance significantly. We conclude that the model properly learns the relation patterns even without injection.

**Boosting techniques:** There are several ways to improve the performance of embedding models: 1) designing a more sophisticated scoring function, 2) proper selection of loss function, 3) using

---

[1]Accordingly, we ran the RotatE code in our setting (embedding dimension 200 and 10 negative samples). The original paper used very big numbers of 1000 and 1000 respectively.

|  | FB15K | | |
| --- | --- | --- | --- |
|  | MR | MRR | Hits@10 |
| RPTransComplEx4(w grounding) | **38** | 72.4 | **88.8** |
| RPTransComplEx4(w/o grounding) | 40 | **72.8** | 88.7 |

Table 4: Link prediction results. RPTransComplEx4(w grounding) is trained on triples and the relation patterns which are grounded. RPTransComplEx4(w/o grounding) is trained on triples and the relation patterns which are not grounded.

more negative samples 4) using negative sampling techniques, 5) enriching dataset (e.g. adding reverse triples). Among the mentioned techniques, we focus on the first and second ones and avoid using other techniques. We keep the setting used in (Trouillon et al., 2016) to have a fair comparison. Using other techniques can further improve the performance of every models including ours. For example, TransComplEx with embedding dimension 200 and 50 negative samples gets 52.2 for Hits@10. Further analyses of our models in a big setting (bigger embedding dimension and more negative samples) are provided in appendix. Still, loss 4 is performing better and we conclude that our theoretical framework is approved in empirical experiments.

## 6 CONCLUSION

In this paper, we reinvestigated the main limitations of Translation-based embedding models from two aspects: *score* and *loss*. We showed that existing theories corresponding to the limitations of the models are inaccurate because the effect of loss functions has been ignored. Accordingly, we presented new theories about the limitations by consideration of the effect of score and loss functions. We proposed TransComplEx, a new variant of TransE which is proven to be less limited comparing to the TransE. The model is trained by using various loss functions on standard dataset including FB15K, FB15K-237, WN18 and WN18RR. According to the experiments, TransComplEx with proper loss function significantly outperformed translation-based embedding models. Moreover, TransComplEx got competitive performance comparing to the state-of-the-art embedding models while it is more efficient in time and memory. The experimental results conformed the presented theories corresponding to the limitations.

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

## A    FURTHER EXPERIMENTS WITH A BIGGER SETTING

In this section we compare TransE, TransComplEx and RotatE trained by using the losses 2 (condition (a),(b)) , 4 (condition (c)) and the RotatE loss (condition (c)). In contrast to our previous experiments, we use a bigger setting: For FB15K-237, we set the embedding dimension to 300 and the number of the negative samples to 256, and for WN18RR, we set the embedding dimension and the number of negative samples to 300 and 250 respectively. We additionally use adversarial negative sampling technique used in (Sun et al., 2019) for all models.

|  | FB15K-237 | | | WN18RR | | |
|---|---|---|---|---|---|---|
|  | MR | MRR | Hits@10 | MR | MRR | Hits@10 |
| TransE2 ($\gamma_1 = 0$) | 222 | 27.4 | 45.7 | 3014 | 19.3 | 47.4 |
| TransE2 ($\gamma_1 > 0$) | 198 | 31.3 | 50.5 | 3942 | 21.4 | 50.3 |
| TransE4 | 181 | 32.3 | 52.1 | 3451 | 23.5 | 53.9 |
| TransE$\mathcal{L}_c^{RotatE}$ | 179 | 32.5 | 51.9 | 3594 | 23.3 | 53.6 |
| TransComplEx2 ($\gamma_1 = 0$) | 213 | 28.5 | 47.3 | 3014 | 31.2 | 49.5 |
| TransComplEx2 ($\gamma_1 > 0$) | 194 | 31.9 | 50.8 | 3942 | 41.3 | 50.8 |
| TransComplEx4 | 177 | 32.8 | 52.1 | 3435 | 44.3 | 55.0 |
| TransComplEx$\mathcal{L}_c^{RotatE}$ | 176 | 32.7 | 51.9 | 3537 | 44.2 | 54.7 |
| RotatE4 | 194 | 33.0 | 52.0 | 3806 | 47.8 | 56.9 |
| RotatE$\mathcal{L}_c^{RotatE}$ | 196 | 33.0 | 51.8 | 3943 | 47.3 | 56.5 |

Table 5: Link prediction results. Rows 1-4: TransE trained using condition (a), (b), (c)(with the loss 4) and (c)(with the RotatE loss) with no injected relation patterns. Rows 5-8 TransComplEx trained using condition (a), (b), (c)(with the loss 4) and (c)(with the RotatE loss) with no injected relation patterns. Rows 9-10: RotatE trained using condition (c)(with the loss 4) and (c)(with the RotatE loss) with no injected relation patterns.

**Analysis of the results**: Table 5 presents a comparison of TransE, TransComplEx and RotatE trained by different losses. TransE2($\gamma_1 = 0$) is trained by using the loss 2 when $\gamma_1 = 0$. TransE2($\gamma_1 > 0$) refers to the TransE model which is trained by using the loss 2 when $\gamma_1$ is a non-zero positive value. The TransE model which is trained by the losses 4 and the RotatE loss (i.e., $\mathcal{L}_c^{RotatE}$) are denoted by TransE4 and TransE$\mathcal{L}_c^{RotatE}$ respectively. The similar notations are considered for TransComplEx and RotatE when they are trained by using different loss functions. The loss 2 with $\gamma_1 = 0$ approximates the condition (a). The loss 2 with $\gamma_1 > 0$ approximates the condition (b). The condition (c) can be approximated by using the loss 4 and the RotatE loss (i.e., $\mathcal{L}_c^{RotatE}$). However, the loss 4 provides a better separation for positive and the negative samples than the RotatE loss. According to the table 5, the loss 4 obtains a better performance than the other losses in each class of the studied models. It is consistent with our theories indicating that the condition (c) is less restrictive. Although we only investigated the main limitations of the translation-based class of embedding models, the theories can be generalized to different models including the RotatE model. From the table, we can see that the loss 4 improves the performance of RotatE. Regarding the table 5, the loss 2 ($\gamma_1 = 0$) gets the worst results. It confirms our theories that with the condition (a), most of the limitations are held. However, with the condition (c), the limitations no longer exist. There have not been any losses that approximate the condition (a). However, most of the theories corresponding to the main limitations of the translation-based class of embedding models have been proven using the condition (a) while the used loss didn't approximate the condition. Therefore, the theories and experimental justifications have not been accurate.

## B    RELATION PATTERN CONVERGENCE ANALYSIS

Figure 4 visualizes the convergence curve of the inverse loss with (RPTransComplEx4) and without (TransComplEx4) injection when the models are trained on WN18. Figure 5 shows the convergence of the TransComplEx4 model trained on FB15K with and without the relation pattern injection. The

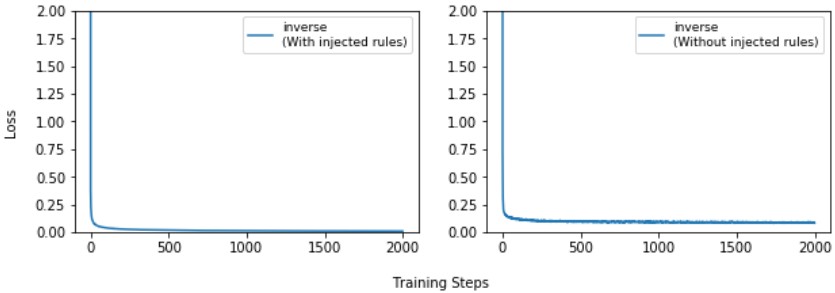

Figure 4: The convergence of inverse relation loss on WN18

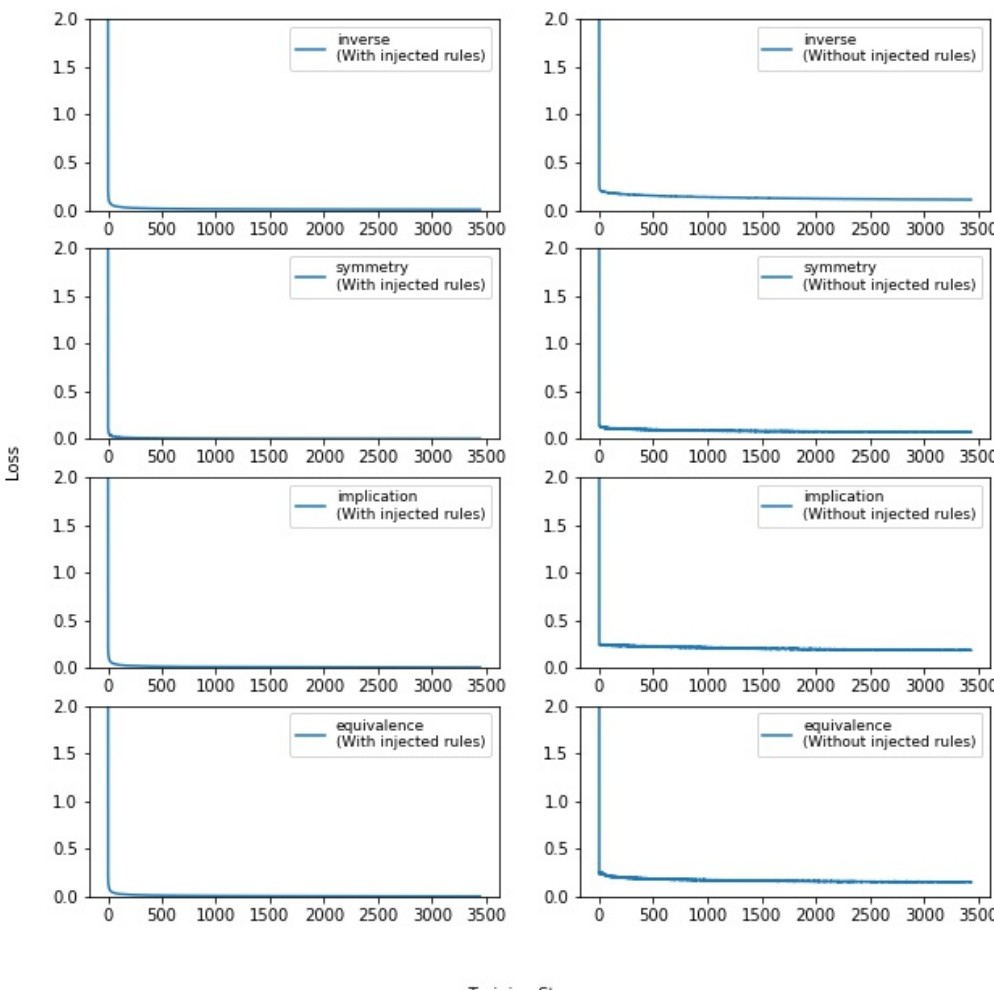

Figure 5: The convergence of inverse, symmetric, implication and equivalence relation losses on FB15K

figures show that the models trained by using the loss 4 can properly encode the relation patterns even without any injection mechanism. In other words, the TransComplEx4 model can properly encode several relation patterns by only training on the triples (without using any additional relation pattern set to be injected). This shows the advantages of the models and the used losses.

## C  PROOF OF THE THEOREM

The proof of theorems are provided as follows:

*Proof of the Theorem T1* 1) Let $r$ be a reflexive relation and condition a) holds. For TransE, we have

$$\mathbf{h} + \mathbf{r} - \mathbf{h} = \mathbf{0}. \tag{9}$$

Therefore, the relation vector collapses to a null vector ($\mathbf{r} = \mathbf{0}$). As a consequence of $\mathbf{r} = \mathbf{0}$, embedding vectors of head and tail entities will be same which is undesired. Therefore, TransE cannot infer reflexive relation with $\mathbf{r} \neq \mathbf{0}$.

For TransComplEx, we have

$$\mathbf{h} + \mathbf{r} - \bar{\mathbf{h}} = \mathbf{0}. \tag{10}$$

We have

$$Re(\mathbf{r}) = \mathbf{0},$$
$$Im(\mathbf{r}) = -2Im(\mathbf{h}). \tag{11}$$

Therefore, all entities will have same embedding vectors which is undesired.

2) Using condition (b), we have

$$\|\mathbf{h} + \mathbf{r} - \mathbf{t}\| = \gamma_1.$$

It gives $\|\mathbf{r}\| = \gamma_1$. Therefore, in order to infer reflexive relation, the length of the relation vector should be $\gamma_1$. Consequently, TransE and TransComplEx can infer reflexive relation. The same procedure can be used for the conditions (c) and (d).

*Proof of the theorem T2:* i) Let the relation $r$ be neither reflexive nor irreflexive and two triples $(e_1, r, e_1), (e_2, r, e_2)$ be positive and negative respectively. Therefore the following inequalities hold:

$$\begin{cases} \|\mathbf{e}_1 + \mathbf{r} - \bar{\mathbf{e}}_1\| \leq \lambda_1, \\ \|\mathbf{e}_2 + \mathbf{r} - \bar{\mathbf{e}}_2\| \geq \lambda_2. \end{cases} \tag{12}$$

Equation 12 is rewritten as follows:

$$\|Re(\mathbf{r}) + i(Im(\mathbf{r}) + 2Im(\mathbf{e}_1))\| \leq \gamma_1,$$
$$\|Re(\mathbf{r}) + i(Im(\mathbf{r}) + 2Im(\mathbf{e}_2))\| \geq \gamma_2, \tag{13}$$

For TransE in real space, $\|Re(\mathbf{r})\| \leq \gamma_1$ and $\|Re(\mathbf{r})\| \geq \gamma_2$ cannot be held simultaneously when $\gamma_2 > \gamma_1$. Therefore, TransE in real space cannot encode a relation which is neither reflexive nor irreflexive. In contrast, TransE in complex space can encode the relation by proper assignment of imaginary parts of entities. Therefore, theoretically TransComplEx can infer a relation which is neither reflexive nor irreflexive.

*Proof of the theorem T3:* i), ii) Let $r$ be a symmetric relation and a) holds. We have

$$\mathbf{h} + \mathbf{r} = \bar{\mathbf{t}},$$
$$\mathbf{t} + \mathbf{r} = \bar{\mathbf{h}}. \tag{14}$$

Trivially, we have

$$
\begin{aligned}
Re(\mathbf{h}) + Re(\mathbf{r}) &= Re(\mathbf{t}), \\
Re(\mathbf{t}) + Re(\mathbf{r}) &= Re(\mathbf{h}), \\
Im(\mathbf{h}) + Im(\mathbf{r}) &= -Im(\mathbf{t}), \\
Im(\mathbf{t}) + Im(\mathbf{r}) &= -Im(\mathbf{h}),
\end{aligned}
\tag{15}
$$

For TransE in real space, there is

$$
\begin{aligned}
Re(\mathbf{h}) + Re(\mathbf{r}) &= Re(\mathbf{t}), \\
Re(\mathbf{t}) + Re(\mathbf{r}) &= Re(\mathbf{h}),
\end{aligned}
$$

Therefore, $Re(\mathbf{r}) = \mathbf{0}$. It means that TransE cannot infer symmetric relations with condition a). For TransComplEx, additionally we have

$$
\begin{aligned}
Im(\mathbf{h}) + Im(\mathbf{r}) &= -Im(\mathbf{t}), \\
Im(\mathbf{t}) + Im(\mathbf{r}) &= -Im(\mathbf{h}),
\end{aligned}
$$

It concludes $Im(\mathbf{h}) + Im(\mathbf{r}) + Im(\mathbf{t}) = \mathbf{0}$. Therefore, TransE in complex space with condition a) can infer symmetric relation. Because a) is an special case of b) and c), TransComplEx can infer symmetric relations in all conditions.

3) For TransE with condition b), there is

$$
\|\mathbf{h} + \mathbf{r} - \mathbf{t}\| = \gamma_1, \tag{16}
$$

$$
\|\mathbf{t} + \mathbf{r} - \mathbf{h}\| = \gamma_1. \tag{17}
$$

The necessity condition for encoding symmetric relation is $\|\mathbf{h} + \mathbf{r} - \mathbf{t}\| = \|\mathbf{t} + \mathbf{r} - \mathbf{h}\|$. This implies $\|h\|cos(\theta_{h,r}) = \|t\|cos(\theta_{t,r})$. Let $h - t = u$, by 17 we have $\|\mathbf{u} + \mathbf{r}\| = \gamma_1$, $\|\mathbf{u} - \mathbf{r}\| = \gamma_1$.

Let $\gamma_1 = \alpha\|r\|$. We have

$$
\begin{cases}
\|\mathbf{u}\|^2 + (1 - \alpha^2)\|\mathbf{r}\|^2 = -2\langle \mathbf{u}, \mathbf{r} \rangle \\
\|\mathbf{u}\|^2 + (1 - \alpha^2)\|\mathbf{r}\|^2 = 2\langle \mathbf{u}, \mathbf{r} \rangle
\end{cases}
\tag{18}
$$

Regarding 18, we have

$\|\mathbf{u}\|^2 + (1 - \alpha^2)\|\mathbf{r}\|^2 = -(\|\mathbf{u}\|^2 + (1 - \alpha^2)\|\mathbf{r}\|^2)$.

$\rightarrow \|\mathbf{u}\|^2 = (\alpha^2 - 1)\|\mathbf{r}\|^2$.

To avoid contradiction, $\alpha \geq 1$. If $\alpha \geq 1$ we have $cos(\theta_{u,r}) = \pi/2$. Therefore, TransE can encode symmetric pattern with condition b), if $\gamma_1 = \alpha\|r\|$ and $\alpha \geq 1$. From the proof of condition b), we conclude that TransE can encode symmetric patterns under conditions c) and d).

*Proof of the theorem T4:* i) The proof of the lemma with condition a) for TransE is mentioned in the paper Kazemi & Poole (2018). For TransComplEx, the proof is trivial. ii) Now, we prove that the limitation L4 is not valid when b) holds.

Let condition b) holds and relation $r$ be reflexive, we have $\|\mathbf{e}_1 + \mathbf{r} - \mathbf{e}_1\| = \gamma_1$, $\|\mathbf{e}_2 + \mathbf{r} - \mathbf{e}_2\| = \gamma_1$.

Let $\|\mathbf{e}_1 + \mathbf{r} - \mathbf{e}_2\| = \gamma_1$. To violate the limitation L4, the triple $(e_2, r, e_1)$ should be negative i.e.,

$\|\mathbf{e}_2 + \mathbf{r} - \mathbf{e}_1\| > \gamma_1$,

$\rightarrow \|\mathbf{e}_2 + \mathbf{r} - \mathbf{e}_1\|^2 > \gamma_1^2$,

$\rightarrow \|\mathbf{e}_2\|^2 + \|\mathbf{e}_1\|^2 + \|\mathbf{r}\|^2 + 2 < \mathbf{e}_2, \mathbf{r} > -2 < \mathbf{e}_2, \mathbf{e}_1 > -2 < \mathbf{e}_1, \mathbf{r} > > \gamma_1^2$.

Considering $\|\mathbf{e}_1 + \mathbf{r} - \mathbf{e}_2\| = \gamma_1$, we have

$< \mathbf{e}_2, \mathbf{r} > - < \mathbf{e}_1, \mathbf{r} > > 0$,
$\rightarrow < \mathbf{e}_2 - \mathbf{e}_1, \mathbf{r} > > 0$,
$\rightarrow cos(\theta_{(\mathbf{e}_2 - \mathbf{e}_1), \mathbf{r}}) > 0$,

Therefore, the limitation L4 is not valid i.e., if a relation $r$ is reflexive, it may not be symmetric. TransE is special case of TransComplEx and also condition b) is special case of condition c). Therefore using conditions b), c) and d), the limitation L4 is not valid for TransE and TransComplEx.

*Proof of the theorem T5*

i) Under condition a), equation $\mathbf{h} + \mathbf{r} - \mathbf{t} = \mathbf{0}$ holds. Therefore, according to the paper Kazemi & Poole (2018), the model has the limitation L5.

ii) If a relation is reflexive, with condition b), we have $\|\mathbf{e}_1 + \mathbf{r} - \mathbf{e}_1\| = \gamma_1, \|\mathbf{e}_2 + \mathbf{r} - \mathbf{e}_2\| = \gamma_1$. Therefore, $\|r\| = \lambda_1$. Let

$$\begin{cases} \|\mathbf{e}_1 + \mathbf{r} - \mathbf{e}_2\| = \gamma_1, \\ \|\mathbf{e}_2 + \mathbf{r} - \mathbf{e}_3\| = \gamma_1. \end{cases} \tag{19}$$

we need to show the following inequality wouldn't give contradiction: $\|\mathbf{e}_2 + \mathbf{r} - \mathbf{e}_3\| > \gamma_1$.

From 19 we have $< \mathbf{e}_2, (\mathbf{e}_1 + \mathbf{e}_2 + \mathbf{e}_3) > < 0$, which is not contradiction.

Therefore, with conditions b) and c), the limitation L5 is not valid for both TransE and TransComplEx.

*Proof of the theorem T6*: i) With condition (a), the limitation L6 is proved in Kazemi & Poole (2018). ii) Considering the assumption of L6 and the condition (b), we have

$$\begin{cases} \|\mathbf{e}_1 + \mathbf{r} - \mathbf{s}_1\| = \gamma_1, \\ \|\mathbf{e}_1 + \mathbf{r} - \mathbf{s}_2\| = \gamma_1. \\ \|\mathbf{e}_2 + \mathbf{r} - \mathbf{s}_1\| = \gamma_1. \end{cases} \tag{20}$$

We show the condition that $\|\mathbf{e}_2 + \mathbf{r} - \mathbf{s}_2\| > \gamma_1$ holds.

Substituting 20 in $\|\mathbf{e}_2 + \mathbf{r} - \mathbf{s}_2\| > \gamma_1$, we have

$cos(\theta_{(s_1-s_2),(e_1-e_2)}) < 0$. Therefore, there are assignments to embeddings of entities that the limitation L6 is not valid with condition (b), (c) and (d).

Figure 6 shows that the limitation L6 is invalid by proper selection of loss function.

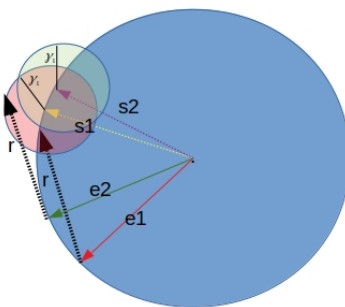

Figure 6: Investigation of L6 with condition (c): The limitation is not valid, because the triple $(e_2, r, s_2)$ can get an score to be considered as negative while triples $((e_1, r, s_1), (e_1, r, s_2), (e_2, r, s_1))$ are positive.

## C.1 FURTHER LIMITATIONS AND FUTURE WORK

In the paper, we have investigated the six limitations of TransE which are resolved by revision of loss function. However, revision of loss functions can resolve further limitations including 1-N, N-1 and M-N relations. More concretely, setting upper-bound for the scores of positive samples can mitigate the M-N problem. We will leave it as future work.

Our theories can be extended to every distance-based embedding models including RotatE etc.

Moreover, the negative likelihood loss has been shown to be effective for training different embedding models including RotatE and TransE. This can also be explained by reformulation of negative

likelihood loss as standard optimization problem, showing the the loss put a boundary for the score functions.

We will consider the mentioned points as future work.

## D    OPTIMAL HYPER-PARAMETERS

The following tables show the optimal configurations for our models included in the Table-2 and 3.

|  | RPTransComplEx2 | | |
| --- | --- | --- | --- |
|  | FB15k | FB15k-237 | WN18 |
| Embedding Dimension ($d$) | 200 | 200 | 200 |
| $\lambda_0$ | 100 | 100 | 100 |
| $\gamma_1$ | 0.4 | 1.5 | 1.0 |
| $\gamma_2$ | 0.5 | 2.0 | 2.0 |
| Negative Sample ($\alpha$) | 10 | 10 | 10 |

Table 6: **Optimal Setting.** The best hyper-parameter setting for RPTransComplEx2 on several dataset.

|  | RPTransComplEx4 | | |
| --- | --- | --- | --- |
|  | FB15k | FB15k-237 | WN18 |
| Embedding Dimension ($d$) | 200 | 200 | 200 |
| $\lambda_0$ | 10 | 100 | 100 |
| $\gamma_1$ | 0.4 | 1.5 | 0.6 |
| $\gamma_2$ | 0.5 | 2.0 | 1.7 |
| Negative Sample ($\alpha$) | 10 | 10 | 2.0 |

Table 7: **Optimal Setting.** The best hyper-parameter setting for RPTransComplEx4 on several dataset.

|  | RPTransComplEx5 | | |
| --- | --- | --- | --- |
|  | FB15k | FB15k-237 | WN18 |
| Embedding Dimension ($d$) | 200 | 200 | 200 |
| $\gamma$ | 5.0 | 10 | 10 |
| Negative Sample ($\alpha$) | 10 | 10 | 10 |

Table 8: **Optimal Setting.** The best hyper-parameter setting for RPTransComplEx5 on several dataset.

|  | TransComplEx4 | | | |
|---|---|---|---|---|
|  | FB15k | FB15k-237 | WN18 | WN18rr |
| Embedding Dimension ($d$) | 200 | 200 | 200 | 200 |
| $\lambda_0$ | 10 | 100 | 100 | 1 |
| $\gamma_1$ | 0.4 | 1.5 | 0.6 | 1.6 |
| $\gamma_2$ | 0.5 | 2.0 | 1.7 | 2.7 |
| Negative Sample ($\alpha$) | 10 | 10 | 2.0 | 2.0 |

Table 9: **Optimal Setting.** The best hyper-parameter setting for TransComplEx4 on several dataset.

|  | TransE 4 | | | |
|---|---|---|---|---|
|  | FB15k | FB15k-237 | WN18 | WN18rr |
| Embedding Dimension ($d$) | 200 | 200 | 200 | 200 |
| $\lambda_0$ | 10 | 100 | 1 | 1 |
| $\gamma_1$ | 0.4 | 0.4 | 1.0 | 0.6 |
| $\gamma_2$ | 0.5 | 0.5 | 2.0 | 1.7 |
| Negative Sample ($\alpha$) | 10 | 10 | 10 | 2 |

Table 10: **Optimal Setting.** The best hyper-parameter setting for TransE4 on several dataset.

