# OpenReview forum: "Toward Understanding The Effect of Loss Function on The Performance of Knowledge Graph Embedding"
_ICLR.cc/2020/Conference — Reject_

### Official Review · AnonReviewer3 · 2019-10-16
**Official Blind Review #3**

**Rating:** 6

**Review:**


Summary:
This paper list several limitations of translational-based Knowledge Graph embedding methods, TransE which have been identified by prior works and have theoretically/empirically shown that all limitations can be addressed by altering the loss function and shifting to Complex domain. The authors propose four variants of loss function which address the limitations and propose a method, RPTransComplEx which utilizes their observations for outperforming several existing Knowledge Graph embedding methods. Overall, the proposed method is well motivated and experimental results have been found to be consistent with the theoretical analysis.

Suggestions/Questions:

1. It would be great if hyperparameters listed in the “Experimental Setup” section could be presented in a table for better readability.

2. In Section 2, the authors have mentioned that RotatE obtains SOTA results using a very large embedding dimension (1000). However, it gives very similar performance even with smaller dimensional embedding (such as 200) with 1000 negative samples. In Section 5, RotatE results with 200 dimension and 10 negative samples are reported for a fair comparison. Wouldn’t it be better to instead increase the number of negative samples in RPTransComplEx instead of decreasing negative samples in RotatE?

3. In Table 3, it is not clear why authors have not reported their performance on the WN18RR dataset for their methods. Also, the reported performance of TransE in [1] is much better than what is reported in the paper.

[1] Sun, Zhiqing, Zhi-Hong Deng, Jian-Yun Nie and Jian Tang. “RotatE: Knowledge Graph Embedding by Relational Rotation in Complex Space.” ArXiv abs/1902.10197 (2019): n. pag.


**Experience Assessment:**

I have published one or two papers in this area.

**Review Assessment: Checking Correctness Of Derivations And Theory:**

I assessed the sensibility of the derivations and theory.

**Review Assessment: Checking Correctness Of Experiments:**

I assessed the sensibility of the experiments.

**Review Assessment: Thoroughness In Paper Reading:**

I read the paper at least twice and used my best judgement in assessing the paper.

---

> ### Author Response · Authors · 2019-11-14
> **Response to Reviewer 3**
>
> Thank you for your precious comments and suggestions.
>
> Comment:"It would be great if hyperparameters listed in the “Experimental Setup” section could be presented in a table for better readability".
>
> Response:
> Thank you for the comments. Following the suggestion, we will put the hyper-parameters in a table in the appendix.
>
> Comment:"In Section 2, the authors have mentioned that RotatE obtains SOTA results using a very large embedding dimension (1000). However, it gives very similar performance even with smaller dimensional embedding (such as 200) with 1000 negative samples. In Section 5, RotatE results with 200 dimension and 10 negative samples are reported for a fair comparison. Wouldn’t it be better to instead increase the number of negative samples in RPTransComplEx instead of decreasing negative samples in RotatE?"
>
> Response:
> Thank you for the suggestion. We decided to use a simple setting to sharply justify our theories. Additionally, we couldn’t use the same RotatE setting due to limitations in the infrastructures.
> However, we additionally ran our experiments using a bigger setting and will include the results in the paper.
> For example, using embedding dimension 300 and 256 negative samples on FB15K-237, RotatE_Hits@10=51.8, TransE5_Hits@10=52.1, TransComplEx5_Hits@10=52.1 and RotatE_MR=195, TransE5_MR=180, TransComplEx5_MR=177. Therefore, using much more negative samples with a bigger embedding dimension, the results improve.
>
> As properly mentioned in the comment of Jingpei Lei, RotatE uses a loss function approximating the condition (c).
> Regarding the RotatE paper, table 13 (appendix), TransE (which was proposed on 2013) trained by the RotatE loss gets a very close performance to the RotatE model (which was proposed on 2019) trained by the RotatE loss when a big setting is used (more negative samples with a higher dimension). A baseline model proposed around six years ago obtains state-of-the-art performance! It has been already reported that the model has several limitations which are not consistent with the recently reported results (e.g., table 13 of the RotatE paper). Our theories corresponding to the limitations of TransE can also explain such reported experimental results and gives a better understanding of the previous work (especially the baselines that several models have been proposed on top of them to fix their limitations). it shows the importance of our work. Overall, TransE and its variants have fewer limitations than what has been reported. Our theories shed more light on this.
>
> Comment:"In Table 3, it is not clear why authors have not reported their performance on the WN18RR dataset for their methods"
>
> Response:
> We didn’t use any relation patterns as extra knowledge to be injected into the model on WN18RR. We only reported the results of TransE5 and TransComplEx5 which are trained by using just triples.
> Comparing TransComplEx and RPTransComplEx on other datasets, we found that TransComplEx encodes relation patterns by training only on triples without any relation pattern injection. We will include figures of convergence of relation pattern losses to show that TransComplEx learns most of the relation patterns without any relation pattern injection.
>
> Comment:"Also, the reported performance of TransE in [1] is much better than what is reported in the paper"
>
> Response:
> [1] reported the result of TransE on WN18RR with different settings (Table5 and 7 of [1]). We decided to rerun experiments on TransE with a Margin Ranking Loss using the setting which we reported in order to have 1) a fair comparison and 2) a proper justification in the theories as the theories are related to TransE with different loss. We will update the results in the paper.

---

### Official Review · AnonReviewer1 · 2019-10-22
**Official Blind Review #1**

**Rating:** 3

**Review:**

In this paper, the authors investigate the main limitations of TransE in the light of loss function. The authors claim that their contributions consist of two parts: 1) proving that the proper selection of loss functions is vital in KGE; 2) proposing a model called TransComplEx. The results show that the proper selection of the loss function can mitigate the limitations of TransX (X=H, D, R, etc) models.

My major concerns are as follows.
1.	The motivation of TransComplEx and why it works are unclear in the paper.
2.	The experiments might be unconvincing. In the experiments, the authors claim that they implement RotatE [1] in their setting to make a fair comparison. However, with their setting, the performance of RotatE is much worse than that in the original paper [1]. Therefore, the experiments might be unfair to RotatE.
3.	There are some typos in this paper. For example, in Line 11 of Section 4.3, the comma should be a period; in Section 5, the "Dissuasion of Results" should be "Discussion of Results".

[1] Zhiqing Sun, Zhi-Hong Deng, Jian-Yun Nie, and Jian Tang. Rotate: Knowledge graph embedding by relational rotation in complex space. arXiv preprint arXiv:1902.10197, 2019.


**Experience Assessment:**

I have read many papers in this area.

**Review Assessment: Checking Correctness Of Derivations And Theory:**

I assessed the sensibility of the derivations and theory.

**Review Assessment: Checking Correctness Of Experiments:**

I carefully checked the experiments.

**Review Assessment: Thoroughness In Paper Reading:**

I read the paper thoroughly.

---

> ### Author Response · Authors · 2019-11-15
> **Response to Reviewer 1**
>
> Thank you for the valuable comments.
>
> Comment:"The motivation of TransComplEx and why it works are unclear in the paper"
>
> Response:
> According to our theories, TransComplEx has fewer limitations than the TransE model.
>
> Comment:"The experiments might be unconvincing. In the experiments, the authors claim that they implement RotatE [1] in their setting to make a fair comparison. However, with their setting, the performance of RotatE is much worse than that in the original paper [1]. Therefore, the experiments might be unfair to RotatE."
>
> Response:
> Comparison of different models in completely different settings does not give a proper conclusion.
> However, following the raised point, we additionally compared the models with RotatE in a bigger setting. We saw the same patterns in the results which confirm our previous conclusions. We will report the results of the new experiments in a bigger setting in the appendix.
>
> Please note that we couldn’t use exactly the same setting of RotatE due to limitations in our infrastructure.
>
> Comment: "There are some typos in this paper. For example, in Line 11 of Section 4.3, the comma should be a period; in Section 5, the "Dissuasion of Results" should be "Discussion of Results"
>
> Response:
> Thank you, we fixed the typos.

---

### Official Review · AnonReviewer2 · 2019-11-03
**Official Blind Review #2**

**Rating:** 1

**Review:**

The paper analyses the effect of different loss functions for TransE and argues that certain limitations of TransE can be mitigated by chosing more appropriate loss functions. Furthermore, the paper proposes TransComplEx -- an adaption of ideas from ComplEx/HolE  to TransE -- to mitigate issues that can not be overcome by a simply chosing a different loss.

Analyzing the behavior and short-comings of commonly-used models can be an important contribution to advance the state-of-the-art. This paper focuses on the performance of TransE, which is a popular representation learning approach for knowledge graph completion and as such fits well into ICLR.

Unfortunately, the current version of the paper seems to have issues regarding methodology and novelty.

Regarding the experimental evaluation: The paper compares the results of TransComplEx and the different loss functions to results that have previously been published in this field (directly, without retraining). However, it seems from Section 5 (Dataset), that this paper is using a modified dataset, as the TransE models are only trained on high-confidence triples. All prior work that I checked doesn't seem to do this, and hence the numbers are not comparable.

Even more serious: Following again Section 5 (Dataset), it seems that the paper imputes all missing triples in the training set for symmetric and transitive relations ("grounding"). Hence,  the models get to see _all_ true triples for these relation types and as such the models in this paper are trained on the test set.

Regarding novelty: The short-comings of TransE and improvements to the loss have been discussed quite extensively in prior work. Using complex representations in TransComplEx seems also a straightforward application of the insights of ComplEx/Hole. As such, the main novelty would lie in the experimental results which, unfortunately, seem problematic.

**Experience Assessment:**

I have published in this field for several years.

**Review Assessment: Checking Correctness Of Derivations And Theory:**

I assessed the sensibility of the derivations and theory.

**Review Assessment: Checking Correctness Of Experiments:**

I assessed the sensibility of the experiments.

**Review Assessment: Thoroughness In Paper Reading:**

I read the paper at least twice and used my best judgement in assessing the paper.

---

> ### Public Comment · ~Jingpei_Lei1 · 2019-11-07
> **About "The short-comings of TransE and improvements to the loss have been discussed quite extensively in prior work. "**
>
> Hello,
>
>       Very sorry for the disturb.
>       I am interested in the limitations of TransE with different loss functions in this paper.
>       Furthermore, I think this is the main contribution of this paper, though I do not know whether the authors of this paper will agree with me or not.
>       First, these "limitations" is not just effect TransE. "Limitations" described in [1] effect all translation-based models, as well as RotatE [2].
>       Second, this paper tell me most of these "limitations" are not real. Since, no one use the lost function in condition a) in training. Most of the translation-based models are trained in  condition d) and the reported best performance of TransE in [2] uses a loss function which can be regarded as a special case of condition c). Most of current papers prove the "limitations"  of TransE and other translation-based models on condition a), while it may be not reasonable enough.
>        I do not think the anaysis of "limitations"  in this paper is very diffcult, but it is really novel to me. Hope to get another related prior work.
>
>       I am a green finger in this scope and can not exhaust all related papers. I have no intention of offending and sorry for the disturb again.
>
> [1]Seyed Mehran Kazemi, David Poole: SimplE Embedding for Link Prediction in Knowledge Graphs. NeurIPS 2018: 4289-4300
> [2]Zhiqing Sun, Zhi-Hong Deng, Jian-Yun Nie, Jian Tang: RotatE: Knowledge Graph Embedding by Relational Rotation in Complex Space. ICLR (Poster) 2019
>
>       BTW, I notice a small error in the last paragraph of proof for Lemma 3 in page 14. But the Lemma 3 in this paper is still correct.
>       "To avoid contradiction, $\alpha \geq 1$.  If $\alpha \geq 1$ we have cos($\theta_{u,r}$) =$\pi$/2".
>       Here,  the "cos($\theta_{u,r}$) =$\pi$/2" is not valid.
>       In fact,  "$\theta_{u,r}$ =$\pi$/2" is a sufficient condition for "$\alpha \geq 1$".   The scope of $\theta_{u,r}$ is depend on $\parallel \textbf{u} \parallel / \parallel \textbf{r} \parallel$.

---

> > ### Author Response · Authors · 2019-11-15
> > **Response to Jingpei Lei**
> >
> > Hello,
> >
> > Thank you for the valuable comments and interest.
> >
> > Comment:"I think this is the main contribution of this paper".
> >
> > Response:
> > We would agree. The main contribution of the paper is re-investigation of the limitations of the translation-based class of embedding models reported in the recent work.
> >
> > Comment: "      First, these "limitations" is not just effect TransE. "Limitations" described in [1] effect all translation-based models, as well as RotatE [2]. "
> >
> > Response:
> > Thanks, We would agree, The theories mentioned in the paper can be generalized to several variants of TransE. We would agree that the theories can be also extended to RotatE. We consider it as future work. However, we provide some experiments in the appendix to show the effect of different loss functions on the performance of the RotatE model.
> >
> > Comment:" Second, this paper tell me most of these "limitations" are not real"
> >
> > Response:
> > That's true. Several new models have been proposed to fix the limitations which didn't really exist.
> >
> > The losses affect the performance of the models. Therefore, they should be considered during the theoretical investigation of the limitations/capabilities of different models. But they have been ignored for a long time. Ignoring the effect of loss causes a big theoretical gap.
> >
> > Comment:" Since, no one use the lost function in condition a) in training"
> >
> > Response:
> > This is an important point. When no one use a loss approximating the condition (a), it is better not to prove theories corresponding the limitations based on this condition. Most of the reported limitations are valid when (a) is enforced by the loss. But the widely used losses do not enforce this condition. We trained the models with the loss approximating the condition (a). The performance reduced. However, most of the widely used losses approximate the condition (c) or (d) in which the reported limitations are not valid.
> >
> > Comment:" reported best performance of TransE in [2] uses a loss function which can be regarded as a special case of condition c)"
> >
> > Response:
> > Thank you for pointing this out. [2] uses condition (a) to investigate the limitation of TransE while during the experiments, the loss function approximates the condition (c). The results of the Table 13 in [2] are consistent with our theories.
> >
> > We present some experiments in the appendix to compare the loss used in RotatE with the loss used in this paper. We saw that the loss used in our paper obtained slightly better performance that the loss used in RotatE. Event it improves the performance of the RotatE model.
> >
> > Comment:"  I do not think the anaysis of "limitations"  in this paper is very diffcult, but it is really novel to me. Hope to get another related prior work"
> >
> > Response:
> > Thank you. The effect of loss function has been ignored for a long time in the theoretical investigations of limitations/capabilities of embedding models while without considering the effect of loss functions, the conclusions are not accurate. This can be clearly seen even in recent works. This shows the novelty and importance of this work.
> >
> > Comment:"      BTW, I notice a small error in the last paragraph of proof for Lemma 3 in page 14. But the Lemma 3 in this paper is still correct."
> >
> > Response:
> > Thank you for the point. $$\theta = \pi/2$$ is the necessity and sufficiency condition considering the equation 9.

---

> ### Author Response · Authors · 2019-11-14
> **Response to Reviewer 2: Regarding the evaluation 1: using modified dataset**
>
> Thank you for the valuable comments.
>
> Comments:" it seems from Section 5 (Dataset), that this paper is using a modified dataset":
>
> Response:
> Actually, we used the same dataset (WN18rr, WN18, FB15K-237, FB15K) that have been extensively used for evaluation of KGEs by others and these data sets do not contain any information about the confidence of triples. Therefore, our models are not trained on high confidence triples. In Section-5 (Dataset), we already mentioned that the relation patterns (i.e., rules) with a lower confidence value are removed. That does not have anything to do with triples and their potential level of confidence. We used the relation patterns (rules) extracted by AMIE. These relation patterns (rules) were used (by doing grounding) in RUGE to be injected into the learning process. Each relation patterns (and not triples) used in RUGE has a confidence value. RUGE also only used relation patterns with confidence higher than 80%. We used the same dataset.
> We compare our models (trained by different losses) with two classes of models: 1) the models that have not used any relation patterns (rules) as background knowledge (such as RotatE and ComplEx, TransE etc), for injection and  2) the models used a set of relation patterns (rules) as background knowledge to inject them into the embedding models during the learning process (such as RUGE, KALE etc). To have a fair comparison, we trained the TransComplEx under two conditions. First: in order to compare with the first class of models, TransComplEx is trained using only triples (Table2,3… TransComplEx row) and we did not use or inject any relation patterns into it. Second: in order to compare with the second class of models, RPTransComplEx used relation patterns (rules) as background knowledge to be injected into the learning process such as RUGE which trained ComplEx using relation patterns with confidence higher than 80%. Therefore, we included both of the models in the Table-2,3 to have a comprehensive evaluation.
> Moreover, comparing TransComplEx5 and RPTransComplEx5 (which both are trained with the same loss function), we see that the results of TransComplEx trained using loss (5) are very close to RPTransComplEx5. We conclude that the model which is trained on only triples with the loss 5 (i.e. TransComplEx5) is capable of properly learning the most of patterns without using additional background knowledge (relation patterns) to be injected. We visualized the relation patterns losses convergence for TransComplEx5 and RPTransComplEx5 (respectively, without and with relation pattern injected). The convergence of the losses confirms that TransComplEx can properly learn the relation patterns without using additional knowledge to be injected. We will include the figures of relation pattern losses convergence of TransComplEx and RPTransComplEx in the paper. We did new experiments on TransComplEx and TransE as well as RotatE with different loss functions with a bigger setting. we will include them in the paper. The results are consistent with the theories corresponding to the limitations of different models. In this experiment, we didn’t use any relation patterns and the models are trained only using triples.

---

> ### Author Response · Authors · 2019-11-14
> **Response to Reviewer 2: Regarding the evaluation 2: Analysis of the results and training on the test**
>
> Thank you for the comment.
>
> Comments:"Even more serious: Following again Section 5 (Dataset), it seems that the paper imputes all missing triples in the training set for symmetric and transitive relations ("grounding"). Hence,  the models get to see _all_ true triples for these relation types and as such the models in this paper are trained on the test set."
>
> Response: Analysis of the results and "train on the test":
> 1: In order to investigate the effect of each of the losses (3, 5, 6), we trained RPTransComplEx3, RPTransComplEx5 and RPTransComplEx6 on the same data (triples and set of relation patterns as background knowledge). From the experiments, we found that the loss 5 (approximating the condition (c) ) obtained a better performance. This is consistent with our theories indicating that the models trained on the condition (c) have fewer limitations.
> Therefore, a comparison of RPTransComplEx3, RPTransComplEx5, and RPTransComplEx6 (which are done on the same data and patterns) concludes that the loss 5 is more effective.
>
> 2: RPTransComplEx5 can be compared with other models injecting relation patterns (use triples and a set of relation patterns) such as RUGE, KALE etc. Please note that RUGE did grounding for rule injection. We used the same data (triples and relation patterns with the confidence of above 80%). Therefore, the comparison is fair because it is done with the same conditions (triples and relation patterns).
>
> 3: We then decided to investigate the performance of RPTransComplEx5 without using additional knowledge (i.e., relation patterns with confidence values) while we already found the loss 5 to be more effective based on theories and previous experiments. Therefore, we trained the TransComplEx with the best-reported loss (i.e. the loss 5 approximating the condition (c)) from the previous experiment, using only triples (not to use relation patterns with confidence and grounding) i.e., TransComplEx5. TransComplEx5 used the same dataset as the first class of embedding models reported in the table 2 and 3 used. The results also confirm the theories.
>
> 4: Given the relation pattern (in the form of Body -> Head), not too many triples in the test set exist in the grounding of Head. For example, our statistics show that only 0.7% (less than one percent) of the test set exist in the grounding of Head for symmetric in FB15K. Therefore, this does not affect our conclusion, especially when all RPTransComplEx# are trained and tested using the same data to conclude that which of the losses is more effective (less restrictive according to theories).
>
> 5:  TransComplEx5 and RPTransComplEx5 obtain close performance. Therefore, the models trained with proper loss function (i.e. 5) encode the patterns properly by only training on triples (i.e., injection of relation patterns with/without grounding didn’t affect the performance significantly). It is further justified when the convergence of the losses of the patterns with and without injection are compared (the relation pattern loss properly converges even without injection). We will include the convergence figures in the paper.
>
> 6: We will report the results of RPTransComplEx5 without the relation patterns which have been grounded to further justify that the high performance is obtained by proper selection of the loss function lifting the limitations. We will additionally include the results of TransComplEx (trained only on triples) with different losses to further support the theories.

---

> ### Author Response · Authors · 2019-11-14
> **Response to Reviewer 2: Regarding the novelty of the paper**
>
> Thank you for the valuable comments.
>
> Comment: "Regarding novelty ..."
>
> Response:
> Regarding the novelty, we believe that this study is not only novel but also has a significant impact on future works as well as better understanding the previous models/losses. Moreover, this study helps to avoid the common mistakes happening in the embedding literature (even in recent works) while the limitations of previous models are investigated.
> There have been a lot of (thousands of) (multi) relational embedding models proposed. During the development of each new embedding model, the researchers have focused on the limitations of previous works to fix them by their proposed model. They have empirically and/or theoretically shown the limitations of previous work and the advantages of their models (capabilities). To the best of our knowledge, many relational embedding papers (including KGEs) have ignored the effect of loss functions during the theoretical investigations of the limitations of previous works and capabilities of their work while without considering the effect of loss functions, the limitations and the theories, as well as experimental evaluations, might not be accurate (in the worst case are incorrect). Ignoring the effect of loss function during the (theoretical) investigations of the limitations of the previous models causes a big (theoretical) gap in the domain of embedding. Several models have been proposed to fix the limitations which did not really exist in the light of our study. Unfortunately, this gap is being enlarged as the effect of the loss function is still being ignored during the investigation of the capability/limitations of new/previous models in the embedding literature (this can be clearly seen in the recent works). We reviewed only a few of the recent works which have ignored the effect of loss functions when they investigate the limitations and prove the capability of their work. We believe that a lot of limitations/capabilities which have been mentioned in the literature for many models should be reinvestigated again as they might be wrong or inaccurate and future works should also consider the effect of loss function when they investigate the limitations of previous works and prove the advantage of their models. The gap is much bigger to be filled by one paper. In our study, we are going to draw attention to the effect of loss function on the limitations of every model (previous ones and future ones). Although we only review some of the limitations of the translation-based class of embedding model in the light of loss function, it can be extended to other classes of embedding models (previous and future work). Without considering loss functions, conclusions may be inaccurate or wrong. The mentioned points show the value and importance of this study.
>
> Regarding “The short-comings of TransE and improvements to the loss have been discussed quite extensively in prior work”, the short-comings of TransE were mis-concluded even in recent work due to ignoring the loss function (e.g., please see the RotatE paper, table-2, symmetric column). Although several loss functions have been proposed so far, most of the works ignore the effect of loss functions while they are highlighting the limitations of previous works and the capability of their model. This indeed causes mis-conclusion.
>
> Comment:"Using complex representations in TransComplEx seems also a straightforward application of the insights of ComplEx/Hole..."
>
> Response:
> The main novelty of this work is to re-investigate the limitations of existing models in the light of loss functions in order to have a more accurate conclusion. Without taking the loss functions into account, the theories of limitations and capabilities are inaccurate. This is the main contribution of this work. TransComplEx is a case study used beside TransE to analyze the main limitations in the light of the loss function.

---

### Official Review · AnonReviewer4 · 2019-11-04
**Official Blind Review #4**

**Rating:** 3

**Review:**

The paper revisits limitations of relation-embedding models by taking losses into account in the derivation of these limitations. They propose and evaluate a new relation encoding (TransComplEx) and show that this encoding can address the limitations previously underlined in the literature when using the right loss.

There seems to be merit in distinguishing the loss when studying relation encoding but I think the paper's analysis lacks proper rigor as-is. A loss minimization won't make equalities in (3) and (5) hold exactly, which the analysis do not account for. A rewriting of the essential elements of the different proofs could make the arguments clearer.

Paper writing:
* The manuscript should be improved with a thorough revision of the style and grammar. Example of mistakes include: extraneous or missing articles, incorrect verbs or tenses.
* The 10-pages length is not beneficial, the recommended 8-pages could hold the same overall content.
* The option list on page 8 is very difficult to read and should be put in a table, e.g. in appendix.
* Parentheses are missing around many citations and equation references

Theory:
Equation (2) and (4) do not seem to bring much compared to the conditions in Table 1. Eq. (3) and (5) show "a" loss function rather than "the" loss function since multiple choices are possible. \gamma_1 should be set to 0 when it is 0 rather than staying in the equations.
* Minimizing the objective (3) and (5) will still not make the conditions in Table 1 hold exactly, because of slack variables.

Experiments:
- Can the authors provide examples of relations learned with RPTransComplEx# that go address the limitations L1...L6, validating experimentally the theoretical claims and showing that the gain with RPTransComplEx5 correspond to having learned these relations?

**Experience Assessment:**

I do not know much about this area.

**Review Assessment: Checking Correctness Of Derivations And Theory:**

I assessed the sensibility of the derivations and theory.

**Review Assessment: Checking Correctness Of Experiments:**

I assessed the sensibility of the experiments.

**Review Assessment: Thoroughness In Paper Reading:**

I read the paper at least twice and used my best judgement in assessing the paper.

---

> ### Author Response · Authors · 2019-11-13
> **Response to Review #4**
>
> Thank you for the valuable comments.
>
> Comment: "A loss minimization won't make equalities in (3) and (5) hold exactly..."
>
> Response:
> In the first step, we posed the conditions (Table-1) to prove theories corresponding to the limitations of the score functions. We then showed that each of these conditions can be approximated by a loss function (which is not unique). The shortcomings of previous works were that they posed limitations under conditions that have not been approximated by the loss function they used. Consequently, they attempted to either address the limitations that do not really exist, or try to show the capability of their model under those conditions that are not approximated by their loss functions, resulting in some of the existing theories and conclusions that are not valid.
> We presented four conditions that defined the region of truth (the region that a triple is considered positive by the model). Under these four conditions, we reinvestigated the main limitations of translation based class of embedding model and posed new theories in this regard.
> In fact, these conditions are ideal conditions to prove theorems and we highlighted that they should be enforced by a proper loss function. In practice, however, such conditions are approximated by a loss function. In other words, we aim to satisfy the condition as much as possible (but not necessarily fully) and as we highlighted, there is not a unique loss function to satisfy them.
> Our work opens a new window for future contributions in the development of new loss functions in which new losses might be proposed to satisfy the conditions, or other existing loss functions/models can be reinvestigated in the light of our proposed conditions. We will add more experimental analysis to the paper (histogram of scores of models trained by different losses, etc) in order to show how each of the losses approximate the corresponding conditions.
>
> Comment: "style and grammar revision "
>
> Response:
> Following your suggestion, we would revise the paper.
>
> Comment:"The 10-pages length is not beneficial ..."
>
> Response:
> We will try to reduce the length of the paper as much as possible.
>
> Comment:"The option list on page 8 is very difficult to read ..."
>
> Response:
> We include the content in a table in the appendix
>
> Comment:"Parentheses are missing around many citations ..."
>
> Response:
> We further revise the paper considering this comment
>
> Comment: "Equation (2) and (4) do not seem to bring much compared to the conditions"
>
> Response:
> We included the conditions of the Table-1 as constraints for the optimizations of (2) and (4). For positive samples, the corresponding constraints enforce conditions to be held. However, for negative samples, we do not intend to include a hard boundary to define the region of negative samples, since some false negative samples are generated during the process of random negative sampling. Instead, we used a soft boundary for the negative samples by adding slack variables to mitigate the negative effect of noise. The goal is to approximate the conditions of Table-1 while addressing the noise in the training data. The optimal solutions (embedding parameters) for the optimization should be in the region defined by constraints. Therefore, the conditions are approximately satisfied.  We experimentally show the value of loss function in order to approximate the conditions.
>
> Comment:" \gamma_1 should be set to 0 when it is 0 rather than staying in the equations."
>
> Response:
> We revise the paper following the comment. We decided to write it in a compact way, that is why we included two conditions in one formulation of loss while mentioning that setting \gamma_1 to 0 approximates the condition (a) and setting \gamma_1 to a positive (non-zero) value approximates the condition (b) for equation (3).
>
> Comment:" Minimizing the objective (3) and (5) will still not make the conditions in Table 1 hold exactly"
>
> Response:
> As we mentioned earlier, the goal is to approximate the ideal conditions that we posed in theory, by a loss function (not to satisfy them strictly). The quality of approximation of each of the conditions by each of the loss function is a new research direction that we are following as future work. Despite this fact, we will add a few results of some of our experiments to show the quality of approximation using the losses. The experiments show that the losses of (3) and (5) are properly converged to very small values, showing that the ideal conditions are properly approximated by the losses.
>
> Comment:"Experiments:  Can the authors provide examples of relations learned  ..."
>
> Response:
> Thank you for the important point. We are running experiments confirming our theories. We will include them in the paper. Our experiments confirm that most of the relation patterns are properly learned by the model (and for some of them even them without injection), showing the value of loss function.

---

### Author Response · Authors · 2019-11-15
**The Revised Version Of The Paper**

We would like to thank the reviewers for their valuable and constructive comments. We uploaded the revised version of the paper addressing the reviewers’ comments and suggestions.

Summary of changes:
1- Revision of the style and grammar (Reviewer 1,4)
2- Inclusion of histogram of the scores of triples to show the losses approximate the conditions (a-d) (Reviewer 4)
3- Training RPTransComplEx without grounding (reviewer 2)
4- Inclusion of the Figures corresponding to the relation pattern loss convergence (reviewer 2,4)
5- Experiments on TransComplEx (without relation pattern injection) with a bigger setting (bigger dimension, more negative samples) are included in the Appendix (reviewer 1,2,3)
6- Moving hyper-parameters in a table to the appendix (reviewer 4, 3)
7- Revision of some parts  of the text to better show the novelty and importance of our work (reviewer 2)

---

### Decision · Program_Chairs · 2019-12-19

**Decision:**

Reject

**Comment:**

The paper analyses the effect of different loss functions for TransE and argues that certain limitations of TransE can be mitigated by choosing more appropriate loss functions.  The submission then proposes TransComplEx to further improve results.  This paper received four reviews, with three recommending rejection, and one recommending weak acceptance.  A main concern was in the clarity of motivating the different models.  Another was in the relatively low performance of RotatE compared with [1], which was raised by multiple reviewers.  The authors provided extensive responses to the concerns raised by the reviewers.  However, at least the implementation of RotatE remains of concern, with the response of the authors indicating "Please note that we couldn’t use exactly the same setting of RotatE due to limitations in our infrastructure."  On the balance, a majority of reviewers felt that the paper was not suitable for publication in its current form.